# ADOPTING DOMAIN-SPECIFIC KNOWLEDGE IN ASR SYSTEMS

**Anton Legchenko & Ivan Bondarenko**
Novosibirsk State University
Novosibirsk, Russia
`{a.legchenko,i.bondarenko}@g.nsu.ru`

## ABSTRACT

This study addresses the challenge of enhancing the accuracy and robustness of multilingual automatic speech recognition (ASR) models in the International Phonetic Alphabet (IPA) format. The primary obstacles include accounting for linguistic diversity, pronunciation variability, and the scarcity of high-quality annotated data for numerous languages, which impedes model generalization to unseen languages. To tackle this issue, we propose a novel approach that integrates prior linguistic knowledge into the training process and incorporates auxiliary information into the model architecture with hierarchical multi-task learning approach. The proposed method decomposes the phoneme recognition process into multiple levels of abstraction, enabling the model to better generalize across diverse phonetic systems. Furthermore, we introduce two variants of language vector representations: one derived from acoustic signals and the other from phonetic transcriptions. These representations serve as auxiliary information, particularly beneficial for few-shot recognition scenarios. We evaluated the approach using datasets that include both high-resource and low-resource languages. The pretrained Wav2vec 2.0 transformer model was employed as the base architecture. As a baseline, the model was fine-tuned solely on the primary task using Connectionist Temporal Classification (CTC) loss, without leveraging auxiliary information. Performance was assessed using Phoneme Error Rate (PER) in both in-domain and out-of-domain scenarios. Experimental results demonstrate that the proposed approach achieves a relative improvement of 7–10% in recognition accuracy across most scenarios. Notably, we observed over 20% improvement for out-of-domain languages when the number of languages in the training dataset was reduced.

## 1 INTRODUCTION

Automatic Speech Recognition (ASR) systems capable of transcribing speech into the International Phonetic Alphabet (IPA) are invaluable tools for various applications, including linguistic research, language education, speech therapy (Cheng et al., 2020) and component ASR systems (Li et al., 2019), (Povey et al., 2011). However, the development of robust and accurate multilingual IPA ASR systems faces significant hurdles. These challenges stem from the inherent linguistic diversity across the world's languages, the substantial variability in pronunciation due to factors like dialect, accent, and individual speaker characteristics (Benzeghiba et al., 2007), and the persistent scarcity of high-quality, annotated data, particularly for low-resource languages (Besacier et al., 2014). This data scarcity significantly hinders the ability of models to generalize effectively to unseen languages.

Current dominant approaches to multilingual ASR often rely on large and diverse datasets, coupled with complex deep learning models, particularly those based on the transformer architecture like Whisper Radford et al. (2023) and Wav2vec 2.0 Baevski et al. (2020). While these methods have achieved impressive results on many tasks, they often struggle with low-resource languages, demonstrating a tendency to overfit to the training data and failing to adequately capture the subtle nuances of diverse phonetic systems. Furthermore, the lack of consistency in annotation practices across different datasets and sources introduces additional complexity and noise into the training data, making

it difficult to train robust and generalizable models. Different annotators may transcribe the same utterance differently, and variations in the level of phonetic detail further compound the issue.

To mitigate these challenges, this study proposes a novel approach that explicitly incorporates prior linguistic knowledge into the training process and leverages auxiliary information within the model architecture. Specifically, we explore the use of Hierarchical Multi-Task Learning (HMTL) (Sanh et al., 2019) to decompose the phoneme recognition process into multiple, hierarchical levels of abstraction. This hierarchical approach allows the model to learn shared phonetic features across languages at higher levels, while simultaneously capturing language-specific phonetic variations at lower levels. We hypothesize that this structured learning process will improve the model's ability to generalize across diverse phonetic systems. Furthermore, we investigate the use of language vector representations, derived from both acoustic signals and phonetic transcriptions, as auxiliary information to enhance model performance, especially in few-shot and zero-shot recognition scenarios where labeled data for a target language is scarce or non-existent.

This research aims to enhance the accuracy and robustness of multilingual IPA ASR systems, ultimately enabling them to generalize more effectively to unseen languages and to better handle the inherent challenges posed by both linguistic diversity and the practical limitations of data scarcity.

## 2 RELATED WORKS

Multilingual ASR has been a subject of extensive research, with various approaches proposed to tackle the challenges of linguistic diversity and data scarcity. Early work in this area often focused on acoustic and language modeling techniques, such as Hidden Markov Models (HMMs) (Rabiner, 1990). However, the advent of deep learning has revolutionized the field, leading to significant improvements in accuracy and generalization.

Deep neural networks, particularly Recurrent Neural Networks (RNNs) (Graves et al., 2013), (Miao et al., 2015) and Convolutional Neural Networks (CNNs) (Collobert et al., 2016), have shown remarkable success in capturing the temporal and spectral characteristics of speech signals. The introduction of the transformer architecture (Vaswani et al., 2017), (Dong et al., 2018) has further propelled the field forward, enabling models to effectively process long sequences and capture long-range dependencies in speech. Models like Wav2Vec 2.0 (Baevski et al., 2020) and Whisper (Radford et al., 2023) have demonstrated impressive performance in multilingual ASR, leveraging self-supervised pre-training on massive amounts of unlabeled data.

In the context of IPA transcription, several studies have explored the use of grapheme-to-phoneme conversion tools like Espeak-ng (esp, 2025) and Phonetisaurus (Novak et al., 2012) to generate phonetic transcriptions from text data. These tools have been used to create large-scale multilingual datasets for training IPA ASR models (Xu et al., 2021). However, the accuracy of these conversion tools can vary across languages, and inconsistencies in annotation practices can introduce noise into the training data.

Multi-task learning (MTL) has emerged as a powerful technique for improving model performance by training a single model on multiple related tasks (Zhang & Yang, 2021). In the context of ASR, MTL has been used to incorporate information from related tasks such as speech activity detection and speaker verification (Sigtia et al., 2020). HMTL was applied to the ASR problem in Sanabria & Metze (2018), where the trained model consistently solves the problem at the phonetic and quasi-morphemic levels.

The use of auxiliary information, such as language embeddings, has also been explored in multilingual ASR (Toshniwal et al., 2018) with training language classification models on speech data. A similar approach, but for speech synthesis, is presented in Lux & Vu (2022). Also Lux & Vu (2022) explores using IPA phoneme classification to improve cross-lingual generalization, based on Mortensen et al. (2016).

## 3 METHODOLOGY

To address the challenges outlined, we propose two distinct approaches and their combinations, all modifying the primary transcription model while maintaining comparability in results. The baseline

model selected for evaluation is the pre-trained Wav2Vec 2.0 XLSR-53 Baevski et al. (2020), which has demonstrated suitability for fine-tuning in multilingual speech recognition tasks Nowakowski et al. (2023), Xu et al. (2021). The baseline approach, adopted from prior works Xu et al. (2021), consists of fine-tuning Wav2Vec 2.0 XLSR-53 on a multilingual dataset transcribed using the International Phonetic Alphabet (IPA). To ensure an objective comparison, both the proposed approaches and the baseline are trained on identical datasets and optimized using the same strategies. The Connectionist Temporal Classification (CTC) loss function Graves et al. (2006) is employed for all models.

The CTC loss function is used to train the model on sequences where the exact alignment between input and output sequences is unknown. The conditional probability for CTC is given by:

$$\mathcal{L}_{\text{CTC}} = -\log P(\mathbf{y}|\mathbf{x}) = -\log \sum_{\mathbf{a} \in \mathcal{A}(\mathbf{x},\mathbf{y})} P(\mathbf{a}|\mathbf{x}),$$

where: - $\mathbf{x}$ is the input speech signal - $\mathbf{y}$ is the target sequence of phonemes (in IPA format); - $\mathbf{a}$ is a possible alignment between $\mathbf{x}$ and $\mathbf{y}$, accounting for possible repetitions and blank symbols; - $\mathcal{A}(\mathbf{x}, \mathbf{y})$ is the set of all possible alignments for the pair $(\mathbf{x}, \mathbf{y})$; - $P(\mathbf{a}|\mathbf{x})$ is the probability of alignment $\mathbf{a}$ given the input $\mathbf{x}$.

The goal of CTC is to maximize the probability of the correct phoneme sequence $\mathbf{y}$ given the input $\mathbf{x}$, summing over all possible alignments.

For training and evaluation, we utilize the Common Voice dataset Ardila et al. (2019), a publicly available multilingual speech corpus, which has been extensively used in related studies Nowakowski et al. (2023), Xu et al. (2021). Since speech-based language vectorization only requires language labels, we leverage all available language samples in the dataset. Speech transcriptions in Common Voice are in the respective native graphemes; therefore, grapheme to phoneme (G2P) tools are employed to convert them into IPA sequences, following the methodology of Xu et al. (2021). Specifically, we adopt the Espeak-ng phoneme conversion tool, though alternative transcription systems can also be utilized within the proposed framework. Model weights are optimized with the AdamW Kingma (2014). Learning rate scheduling follows a cyclic strategy Smith (2017), which has been shown to enhance the fine-tuning of pre-trained models. Mini-batch training Wilson & Martinez (2003) is applied throughout all experiments. The evaluation metric used for phoneme recognition is Phoneme Error Rate (PER), computed as the Levenshtein distance between predicted and reference phoneme sequences, excluding delimiters. Individual symbols in the phoneme representation were considered, resulting in a smaller penalty for errors involving similar phonemes, such as those differing only in diacritics. The PER was calculated using the following formula:

$$\text{PER} = \frac{S + D + I}{N} \times 100\%, \tag{1}$$

where $S$ is the number of substitutions, $D$ is the number of deletions, $I$ is the number of insertions, and $N$ is the total number of phonemes in the reference transcription.

## 3.1 HIERARCHICAL MULTI-TASK LEARNING FOR PHONEME RECOGNITION

We introduce a hierarchical multi-task learning (HMTL) pipeline that leverages the phoneme classification schema defined by IPA to group acoustically similar phonemes. These phoneme groups serve as higher-level abstractions, forming an intermediate learning objective within the hierarchical model. Specifically, phonemes are replaced with their respective class labels according to IPA consonant and vowel classification tables. The resulting sequence of class symbols acts as an auxiliary prediction target. To implement this, we introduce an additional classification head, structurally identical to the primary phoneme classifier, which optimizes an auxiliary CTC loss. Multiple phoneme groupings are explored, including classifications based on IPA table rows, columns, and their combinations. The model is trained jointly on both tasks at each optimization step, with the final loss function comprising the sum of both task-specific losses. The loss function for the hierarchical multi-task learning framework is defined as:

$$\mathcal{L}_{\text{HMTL}} = \mathcal{L}_{\text{CTC}}^{\text{phoneme}} + \lambda \cdot \mathcal{L}_{\text{CTC}}^{\text{class}}, \tag{2}$$

where $\mathcal{L}_{\text{CTC}}^{\text{phoneme}}$ is the CTC loss for the primary phoneme recognition task, as defined earlier; $\mathcal{L}_{\text{CTC}}^{\text{class}}$ is the CTC loss for the auxiliary task of predicting phoneme classes, which are simplified representations of the phoneme sequences; and $\lambda$ is a weighting hyperparameter that balances the contribution of the auxiliary task to the total loss.

The auxiliary task $\mathcal{L}_{\text{CTC}}^{\text{class}}$ operates on a simplified version of the phoneme sequence, where each phoneme is replaced by its corresponding class label (e.g., based on IPA consonant and vowel categories). This simplification reduces the complexity of the sequence, allowing the model to learn higher-level phonetic patterns that are shared across languages. The choice of phoneme classes is guided by the IPA classification schema, which groups phonemes based on shared articulatory or acoustic features. The overall training scheme is depicted in Figure 1.

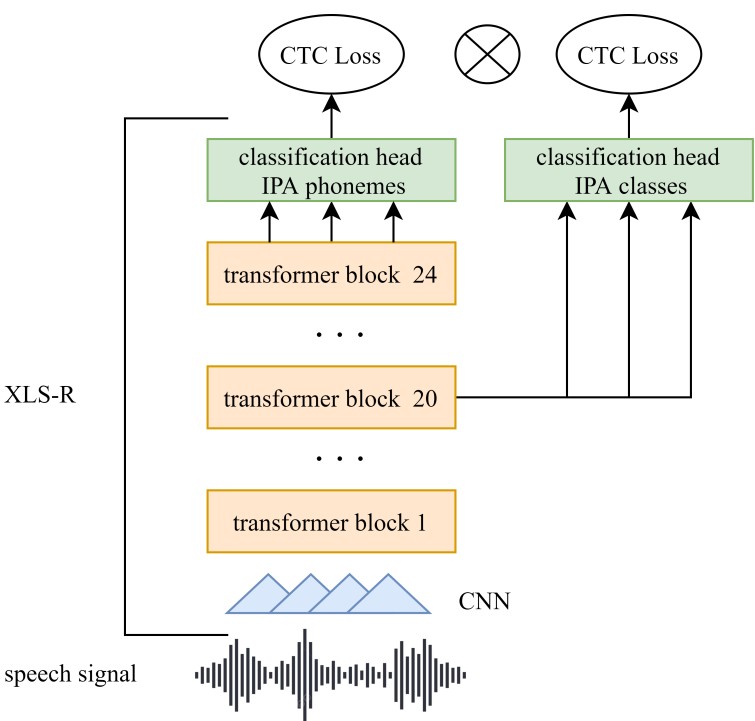

Figure 1: The Training scheme of the hierarchical multi-task model

The advantages of HMTL for multilingual phoneme recognition can be hypothesized as follows. First, joint training across multiple abstraction levels may facilitate improved generalization across diverse phonetic systems, as the intermediate phoneme-group representations potentially capture shared phonetic patterns common to multiple languages. Second, we hypothesize that the model's robustness to pronunciation variability, background noise, and annotation inconsistencies could be enhanced by abstracting phonemes into broader phonetic groups. Such generalized phoneme classes are expected to provide more consistent and reliable training signals, particularly in cases of noisy audio data or ambiguous and erroneous labels. Finally, it is plausible that the hierarchical structure enables the model to leverage phonetic commonalities across languages, potentially leading to better performance in low-resource scenarios by transferring shared acoustic representations.

### 3.2 Auxiliary language embedding models

Incorporating auxiliary linguistic information can improve transcription accuracy both for seen languages in the training dataset, as demonstrated in Toshniwal et al. (2018), and for unseen languages, by transferring knowledge from related languages. To achieve this, we introduce language vector representations derived from both speech and phonetic transcriptions, utilizing metric learning techniques.

#### 3.2.1 Speech-based embedding models

For speech-based language embeddings, we employ Wav2Vec 2.0 XLSR-53, leveraging its feature extraction capabilities. Metric learning is implemented using the triplet loss function Hoffer & Ailon (2015), where triplets consist of two samples from the same language and a negative sample from a different, randomly selected language. The triplet loss is defined as:

$$\mathcal{L}_{\text{triplet}} = \max\left(0, d(\mathbf{e}_{\text{anchor}}, \mathbf{e}_{\text{positive}}) - d(\mathbf{e}_{\text{anchor}}, \mathbf{e}_{\text{negative}}) + \alpha\right), \tag{3}$$

where $\mathbf{e}_{\text{anchor}}$ is the embedding of the anchor sample; $\mathbf{e}_{\text{positive}}$ is the embedding of a positive sample, which belongs to the same language as the anchor; $\mathbf{e}_{\text{negative}}$ is the embedding of a negative sample, which belongs to a different language; $d(\cdot, \cdot)$ is a distance function (e.g., Euclidean or cosine distance); and $\alpha$ is a margin parameter that ensures a minimum separation between the distances to positive and negative samples.

The triplet loss aims to minimize the distance between the anchor and positive samples while maximizing the distance between the anchor and negative samples, thereby improving the discriminative power of language embeddings.

While hard negative sampling strategies (e.g., selecting a related language as the negative sample) can further improve quality of final embeddings, we do not employ them in this study. The XVector approach Snyder et al. (2019) is used to generate a fixed-length vector representation for each speech sequence, and language-level representations are obtained by averaging embeddings with mean-pooling across all available samples for a given language.

#### 3.2.2 Transcription based embedding models

For transcription-based language embeddings, we utilize PhoneBERT Li et al. (2023), a BERT-based model pre-trained using masked language modeling (MLM) Devlin (2018) on phonetic transcriptions from over 100 languages. Sequence-level representations are obtained by mean-pooling the output embeddings across all phoneme tokens in a transcription, mirroring the approach used for speech-based embeddings. The general workflow of our language embedding approach is illustrated in Figure 2.

Integrating auxiliary language vector representations into the transcription model has the potential to significantly improve phoneme recognition, particularly in zero-shot scenarios. When limited speech recordings or phonetic transcriptions from an unseen language are available, language embeddings can be computed and fed into the transcription model as additional context, facilitating adaptation to new linguistic patterns and improving phoneme sequence prediction accuracy.

#### 3.2.3 Evaluation of language embedding models

To assess the quality of the proposed language vectorization models, we employ the following evaluation protocol:

1. Extract vector representations for test samples from either phonetic transcriptions or speech recordings.
2. Randomly split the vectorized samples into training and test subsets according to a predefined ratio.
3. Train a multi-class logistic regression classifier (Bottou, 2010) using stochastic gradient descent on the language classification task, leveraging the vector representations as input features.

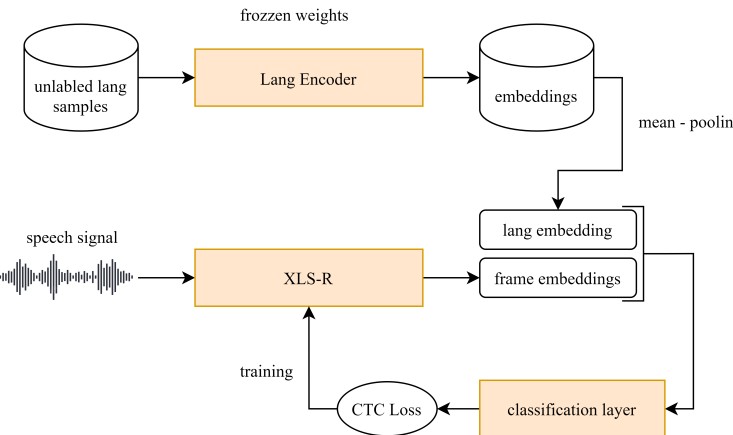

Figure 2: Inference and training scheme with language auxiliary information based on available language samples

    4. Evaluate the classifier on the test set and compute the classification accuracy.

This evaluation pipeline provides an effective means to quantify the informativeness of language embeddings for distinguishing between different languages. Higher classification accuracy indicates better separation of language-specific features, which, in turn, enhances the utility of these representations in multilingual phoneme recognition tasks.

## 4 EXPERIMENTAL RESULTS

The training and testing datasets were structured as follows:

- Each language was limited to a total of 10 hours of speech for training and 2 hours for testing, if the language participated in the training set. If a language was held-out, 10 hours of speech were used for testing on that language.
- Only recordings without negative ratings and with at least one positive rating were used.
- Common Voice version 16.0 was utilized.

Experiments were conducted using subsets with varying numbers of languages: 5, 9, 13, and 27. Their composition is detailed in Table 1.

Table 1: List of languages used in various experiments

| Number of experiment languages | Language codes |
|---|---|
| 5 | af, ca, es, de, fr |
| 9 | af, as, ca, de, es, et, fa, hi, fr |
| 13 | af, as, ca, de, es, et, fa, hi, hu, hy-AM, id, ja, fr |
| 27 | af, as, ca, de, es, et, fa, hi, hu, hy-AM, id, ja, lt, mk, ne-NP, or, pt, ro, ru, sk, sr, sv-SE, ta, tr, ur, vi, fr |

The transcription-based language embedding model was trained on a subset of two million examples from Deri & Knight (2016). For readability of the result tables of intermediate experiments, obtained values are presented for the case where the held-out language was French (fr). Results for other splits are shown in final results - table 5.

### 4.1 HIERARCHICAL MULTI-TASK MODEL

A series of experiments was conducted on various datasets. Furthermore, on the largest dataset with 12 training languages, a selection of auxiliary head positions and IPA phoneme classifications was

performed. The results of these selection experiments are presented in Table 2. The evaluation results indicate the best PER (Phone Error Rate) scores achieved during training, as well as the metrics recorded throughout the training process. Training was halted using either classic early stopping (ES) (Prechelt, 2002) on in-domain held-out data or cross-validation (CV) (Kohavi, 1995) with training languages partitioned into folds.

Table 2: PER metrics on test set for 12 training languages, depending on HMTL configuration

| Layer of additional heads / IPA consonants / IPA vowels | PER out of domain | | PER in domain | |
|---|---|---|---|---|
| | ES | CV | ES | CV |
| Baseline | 0.358 | 0.348 | 0.113 | 0.117 |
| 16 / Columns / Columns | 0.331 | 0.271 | 0.107 | 0.115 |
| 20 / Columns / Columns | **0.323** | **0.264** | **0.103** | **0.109** |
| 24 / Columns / Columns | 0.345 | 0.294 | 0.111 | 0.116 |
| 16 / Rows / Rows | 0.339 | 0.279 | 0.111 | 0.115 |
| 20 / Rows / Rows | 0.334 | 0.270 | 0.108 | 0.112 |
| 24 / Rows / Rows | 0.341 | 0.292 | 0.109 | 0.116 |
| 20 / Columns / Rows | 0.329 | 0.267 | 0.105 | 0.110 |
| 20 / Rows / Columns | 0.337 | 0.282 | 0.110 | 0.115 |

The obtained results show that the best location for the additional transcription head is one of the final layers of the model, different from the output layer. The fact that the best location for the additional transcription head on the 20th layer of the main model, compared to the location with the final classification head, also indicates that the improvement in results compared to the simple solution is not only due to a change in the loss function, but also due to model's need to learn to solve the task hierarchically, starting from a simpler subtask.

After fixing the configuration for representing the IPA-based auxiliary data and fixing the model architecture, a series of experiments was conducted to determine the approach's dependence on the number of languages represented in the training sample. The results are presented in Table 3.

Table 3: PER metrics on test set for 4, 8, and 12 training languages

| Approach | PER out of domain | | PER in domain | |
|---|---|---|---|---|
| | ES | CV | ES | CV |
| **4 Training Languages** | | | | |
| Baseline | 0.447 | 0.421 | 0.151 | 0.153 |
| HMTL | 0.396 | 0.365 | 0.142 | 0.147 |
| **8 Training Languages** | | | | |
| Baseline | 0.386 | 0.374 | 0.132 | 0.138 |
| HMTL | 0.316 | 0.287 | 0.111 | 0.121 |
| **12 Training Languages** | | | | |
| Baseline | 0.358 | 0.348 | 0.113 | 0.117 |
| HMTL | 0.323 | 0.264 | 0.103 | 0.109 |

On all samples, the model showed a good improvement in quality on both the external and internal tests, at the level of 10-20% PER reduction after establishing training, and more than 30% when comparing the best metrics during training on the external PER.

Then, the possibility of using two additional classification heads, implementing various IPA classification representations, was tested. A configuration with the heads located on the same layer was

considered, since in the IPA itself these classification levels are not hierarchical, but form a table. The optimal configuration obtained earlier was used as the first auxiliary head (the head is located on the 20th layer of the model, and columns of the IPA are used for both consonants and vowels), the auxiliary head used the same location and the opposite set of classifications accordingly. Unfortunately, the second additional transcription head noticeably worsened the results, which is most likely due to the model's need to solve an almost final task on the preliminary layers, since after predicting both the column and the row, in most cases only one or two phonemes remain to be selected.

## 4.2 MODELS WITH AUXILIARY LANGUAGE INFORMATION

### 4.2.1 SPEECH-BASED EMBEDDING MODEL

As part of the study on the possibility of introducing auxiliary information about the language of the recognized speech, a speech signal vectorization model was trained based on the Wav2vec 2.0 XLSR-53 model Baevski et al. (2020) with 300 million parameters. The difference from main transcription model include existence of the head feature for extraction based on XVector Snyder et al. (2019), implementing obtaining a single vector of dimension of 512 features for the entire speech signal, in contrast to the phoneme prediction head. The model was trained using triplet loss metric-learning, triplets were formed based on the transcription's language. Training was carried out using a stopping criterion based on metrics on a held-out sample represented by examples in 4 languages, and examples belonging to 20 other languages were used for training. The model showed good trainability and reached convergence by the 3rd epoch, reaching a metric of 0.88.

For a clear visualization of the results, the UMAP method McInnes et al. (2018) was used to reduce the dimensionality of the obtained vector representations to 2D, making the data convenient to visualize on a plane. Fig. 3 shows these representations of speech signals from languages in the test sample projected into two-dimensional space.

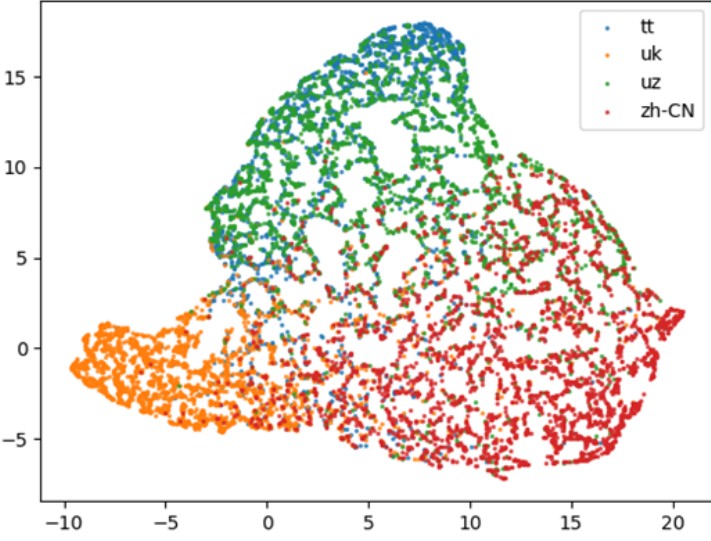

Figure 3: Visualization of the two-dimensional UMAP projection of test data embeddings

As can be seen from the visual representation of the data, even in a space of dimension two, uk and zh-CN are clearly well separated, while tt and uz have significant overlap, corresponding to the actual relationship between the Tatar (tt) and Uzbek (uz) languages, both languages belong to the Turkic group of languages.

### 4.2.2 TRANSCRIPTION-BASED EMBEDDING MODEL

As part of the study on the possibility of introducing auxiliary information about the language of the recognized speech, a embedding model was trained based on the text representation of IPA

transcriptions, obtained using the Espeak-ng phonetization algorithm applied to texts in more than 100 languages and a total number of transcriptions exceeding one million. Data from the work by Deri & Knight (2016) was used for low-resource languages. Training control and metric tracking were carried out similarly to the speech-based model, in this case 15 languages were set aside for testing. The basis for training is a BERT-type model trained using by masked language modeling (MLM) Devlin (2018) on phonetic transcriptions as part of research Li et al. (2023). BERT is used to obtain a single vectoral display The transcription's method simply averages all the token embeddings in the sequence obtained from BERT. Similar to speech-based approach, training was carried out using the Triplet Loss, triplets were formed based on the transcription's language. The model reached convergence after two epochs of training, reaching an average accuracy level of 0.917 on the held-out languages.

### 4.2.3 ASR MODEL WITH ADDITIONAL LANGUAGE SPEECH-BASED EMBEDDINGS

To verify the hypothesis put forward, a series of experiments were conducted by varying the number of languages participating in the experiment, as well as their distribution between the training and test samples. When conducting experiments, the test sample was represented by two parts: 20% of the data set aside for languages present in training, as well as data from two held-out languages not represented in the training sample. As the vector transmitted further, we tried using both the vector from the audio recording itself and the averaged vector from all available speech examples in the recognizable language. The second option showed noticeably better results and only this method will be considered further. An additional experiment was also conducted to check the significance of the transmitted information about languages, to exclude the increase in metrics solely due to the transmission of language labels. The additional experiment consisted of training the model in a similar way, but with the auxiliary embeddings replaced with one-hot vectors, reflecting belonging to one of the languages. When testing on a new language, all positions were replaced with 0s. The results of the experiment are also presented in Table 4.

Table 4: Quality Estimates on External and Internal Test for 4, 8, and 12 Training Languages

| Approach | PER out of domain | | PER in domain | |
|---|---|---|---|---|
| | ES | CV | ES | CV |
| **4 Training Languages** | | | | |
| Baseline | 0.447 | 0.421 | 0.153 | 0.151 |
| Self additional embeddings | 0.431 | 0.418 | 0.142 | **0.147** |
| Averaged additional embeddings | **0.421** | **0.405** | **0.147** | 0.148 |
| **8 Training Languages** | | | | |
| Baseline | 0.386 | 0.374 | 0.132 | 0.138 |
| Self additional embeddings | 0.361 | 0.361 | 0.125 | **0.127** |
| Averaged additional embeddings | **0.357** | **0.348** | **0.127** | 0.128 |
| **12 Training Languages** | | | | |
| Baseline | 0.358 | 0.348 | 0.113 | 0.117 |
| Self additional embeddings | 0.325 | 0.315 | 0.102 | 0.107 |
| Averaged additional embeddings | **0.321** | **0.301** | **0.104** | **0.106** |
| One-hot language embeddings | 0.361 | 0.351 | 0.112 | 0.115 |

The results indicate a logical conclusion: more complete information in the case of individual embeddings allows the model to build more complex dependencies based on the data being studied, which in turn reduces the generalizing ability to new languages, compared to using only averaged information about languages.

Transmission of only one-hot vectors, indicating to the model the language being recognized, did indeed somewhat improve the result compared to the simple solution on the internal test, but the increase is significantly less than from using specialized trained embeddings, while on the external test a deterioration in results was expected. The results obtained show the achievement of the method

for improving target metrics, and also show the significance of using the proposed trained language representations.

Another important observation from the results obtained is the increased efficiency of approaches from data scaling. The larger the sample of training languages, the greater the increase compared to the simple solution was from using the approach, related to the ability to better generalize the use of auxiliary information during transcription.

### 4.2.4 ASR MODEL WITH ADDITIONAL LANGUAGE TRANSCRIPTION-BASED EMBEDDINGS

The conditions for conducting experiments to test the hypothesis were similar to the method using speech signals. Similar patterns were obtained as in the previous section, and therefore we present only data from an experiment with 12 training languages - Table 5.

The results shows the performance increase on target metric from using approach on both internal and external test. It show results is better than solution based one One-hot vectors.

### 4.3 COMBINING APPROACHES AND FINAL RESULTS

The next stage of the study was to test the compatibility of the previously proposed approaches with each other. The obtained results for a dataset of 12 trained languages are presented below in tables 15 and 16. For comparison, the table also presents the simple solution result and the best metric previously obtained by the hierarchical model.

Table 5: Best PER on held-out languages: French – fr, German – de, Catalan – ca, Japanese – ja.

| Approach | PER in-domain | PER fr | PER de | PER ca | PER ja |
|:---:|:---:|:---:|:---:|:---:|:---:|
| **12 Training Languages** | | | | | |
| Baseline | 0.113 | 0.348 | 0.314 | 0.361 | 0.423 |
| HMTL | 0.103 | 0.264 | 0.253 | 0.323 | 0.392 |
| Speech embeddings | 0.106 | 0.301 | 0.281 | 0.333 | 0.391 |
| Transcription embeddings | 0.103 | 0.321 | 0.279 | 0.337 | 0.385 |
| Speech embeddings + transcription embeddings | 0.098 | 0.311 | 0.275 | 0.324 | 0.373 |
| Transcription embeddings + HMTL | 0.092 | 0.261 | 0.271 | 0.339 | 0.378 |
| Speech embeddings + HMTL | 0.091 | 0.252 | 0.276 | 0.341 | 0.372 |
| Combination of all methods | 0.091 | 0.249 | 0.245 | 0.325 | 0.361 |
| **26 Training Languages** | | | | | |
| Baseline | 0.93 | 0.265 | 0.249 | 0.311 | 0.323 |
| HMTL | 0.089 | 0.249 | 0.238 | 0.301 | 0.315 |
| Combination of all methods | 0.088 | 0.251 | 0.234 | 0.292 | 0.281 |

The focus of this study is on the out-of-domain scenario, providing a detailed comparison for four different held-out languages, as shown in Table 5. The results indicate that all proposed approaches significantly outperform the baseline. For the in-domain PER, the metric increase ranges from 9 to 15%, while for the out of domain, we get an increase from 8 to 30%. In general, the models has different ratio in domain /out of domain metrics, need approach selection based based on objective, improve quality already language presented by training or make model more resistant to new language.

Combining approaches to target goal increase metrics, combined approaches result get over baseline, even from result approach separated, 30% result out of domain data baseline model with approach, with data training and 7-10% best of baseline solution approaches by separated.

As demonstrated by the conducted experiments, expanding the training dataset by increasing the number of languages included improves the performance of all methods. At the same time, the

approaches proposed in this work still yield a significant improvement over the baseline solution, achieving a relative PER improvement of 7 to 8%.

## 4.4 LANGUAGE REPRESENTATION SUBSTITUTION

To better understand the impact of language vector representations on the performance of the combined approach, we conducted an additional experiment. Specifically, we replaced the language embedding of a held-out language with embedding from other languages during testing. For this experiment, German (de) was selected as the held-out language, and its vector was substituted with embeddings from English (en), Swedish (sv-SE), Russian (ru), and Japanese (ja). In particular, all languages, except English, were included in the training dataset. The results of this experiment are presented in Table 6.

Table 6: PER Metrics on German (de) with substituted language embeddings

| Language of embedding | PER de |
|---|---|
| Baseline | 0.249 |
| German (de) | 0.234 |
| English (en) | 0.243 |
| Swedish (sv-SE) | **0.225** |
| Russian (ru) | 0.295 |
| Japanese (ja) | 0.301 |

The results demonstrate that using embedding from closely related languages within the Germanic language family—such as Swedish and English—yielded competitive performance. Notably, the use of Swedish embedding achieved the best results, likely because the model was already familiar with this embedding during training and could process it more effectively. Conversely, substituting embeddings from languages belonging to different language families, such as Russian and Japanese, led to a significant degradation in transcription accuracy, resulting in performance worse than the baseline.

From a practical perspective, this experiment highlights the potential utility of leveraging vector representations from linguistically similar languages, particularly those sharing phonetic or genealogical characteristics. This approach is especially valuable when high-quality data for generating language-specific embedding is unavailable. For instance, in scenarios where a target language lacks sufficient speech or transcription data, embeddings from a closely related language could serve as a viable substitute, enabling the model to maintain reasonable performance.

## 5 CONCLUSION

This work investigated approaches to enhance the generalization capabilities of multilingual Automatic Speech Recognition (ASR) systems in the International Phonetic Alphabet (IPA) format. The proposed methods demonstrated significant improvements in recognition accuracy for both in-domain and out-of-domain languages. The results align with established linguistic knowledge regarding language relatedness, validating the effectiveness of the proposed approaches. The experiments yielded meaningful results, showing a relative improvement of 7–10% in recognition accuracy across most scenarios, with over 20% improvement observed for out-of-domain languages when the number of training languages was reduced. The final contribution of this work includes training all components on a broader dataset and making the model publicly available. Future research directions include a more detailed exploration of the relationships between languages in the learned vector spaces and their impact on transcription accuracy, as well as comparisons with linguistic theories. Additionally, the creation of a unified vector space that integrates both acoustic and phonetic modalities, reflecting the transformation principles from speech to transcription for specific languages, presents an intriguing avenue for further investigation.

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

## A    EXPERIMENTAL SETUP

This appendix details the computational resources and training specifics relevant to the experiments conducted in this study. All experiments were performed utilizing *NVIDIA Tesla A100 80GB* and *NVIDIA RTX 3090Ti 24 GB* GPUs. With respect to the datasets and model architectures employed in this work, the training time for a single epoch was approximately one hour per one A100 GPU for a set of 12 languages and proportionally to the number of languages for the remaining experiments. Furthermore, the incorporation of auxiliary loss functions introduced a moderate computational overhead, resulting in an approximate increase in training time of within 20%. In most experiments with comparing HMTL and baseline, the optimal number of epochs when using cross-validation for out-of-domain scenario ranged from 1 to 2 epochs, while for the baseline, it was similar but slightly higher, at 2–3 epochs. However, no practically significant or predictable results were obtained, making it difficult to discuss the results in terms of convergence acceleration. This issue will be explored in greater detail in future work.

