# OpenReview forum: "Adopting Domain-Specific Knowledge in ASR Systems"
_mathai.club/MathAI/2025/Conference — MathAI 2025 Oral_

### Official Review · Reviewer_tFRQ · 2025-02-24
**Good research but sloppy paper**

**Rating:** 7
**Confidence:** 4

**Review:**

The paper proposes a multitask learning approach for the multilingual automatic speech recognition (ASR) using International Phonetic Alphabet (IPA). Specifically, the authors propose additional task of learning phoneme class at a deeper layer. The paper also performs experiments on two representations of a language vector. The approach proposed provides meaningful improvements over the baselines. The research is in line with a current trend to train speech-related models using multitask approach.

General question: What IPA classification produced the best result? How about an ablation of table columns vs. table rows?

Overall, the research is well planned and performed, and most of the ablations required are conducted. Still, the paper itself is written highly sloppily which significantly degrades the paper impact and needs to be fixed. Things to be fixed:

The paper title is misleading. Domain-specific knowledge for an ASR model may refer to acoustic or linguistic domain.

Citations: cite like on line 395, not like on line 391

line 16 Change “languages.Tackle” to  “languages. To tackle”

line 17 - 18 “incorporates auxiliary information into the model architecture“ is an incorrect claim. The model architecture does not change.

line 29 - 30 “Experimental results demonstrate that the proposed approach achieves a relative improvement of 7–10” – sentence is unfinished.

line 132 - 136 This looks like near-duplicate with an editing artifact “The evaluation metric used for phoneme recognition is Phoneme Error Rate (PER), computed as the Levenshtein distance between predicted and reference phoneme sequences, excluding delimiters. he Phone Error Rate (PER) metric, based on the Levenshtein distance applied to phonemes (excluding separators), was used to evaluate transcription quality.”

line 155 Remove duplicated  “The model is trained The model is trained”

line 170 no ablations are ever given for preferring block 20 vs. 21 or 19

lines 199 - 205 ablations are needed for each claim

lines 391 - 395 Please deduplicate two sentences.

lines 417 - 420 two sentences contradict each other

line 479 Something is missing: “embeddings. a, while on the external”

lines 529 - 531 cannot be decoded. Rewrite.

lines 533 – 535 rewrite for clarity

---

### Official Review · Reviewer_FQiC · 2025-02-25
**Overall, the paper presents a well-designed and innovative approach to improving multilingual ASR systems, particularly in the context of IPA transcription. The integration of linguistic knowledge and hierarchical multi-task learning, along with the use of language vector representations, represents a significant contribution to the field. However, the paper would benefit from a more comprehensive comparison with state-of-the-art models, a discussion of computational costs, and more detailed experiments on zero-shot learning. With these improvements, the paper would provide a more complete and compelling case for the proposed methods.**

**Rating:** 9
**Confidence:** 2

**Review:**

The paper addresses the challenge of improving the accuracy and robustness of multilingual Automatic Speech Recognition (ASR) systems, particularly in the context of transcribing speech into the International Phonetic Alphabet (IPA). The authors propose a novel approach that integrates prior linguistic knowledge into the training process and employs hierarchical multi-task learning (HMTL) to decompose the phoneme recognition process into multiple levels of abstraction. Additionally, the paper introduces two variants of language vector representations—derived from acoustic signals and phonetic transcriptions—to enhance few-shot and zero-shot recognition scenarios. The proposed methods are evaluated using datasets that include both high-resource and low-resource languages, with the Wav2vec 2.0 transformer model serving as the base architecture. The results demonstrate significant improvements in Phoneme Error Rate (PER), particularly for out-of-domain languages, with relative improvements ranging from 7% to 20-30%.

Strengths:

Contribution: The paper presents a novel approach to multilingual ASR by integrating linguistic knowledge and hierarchical multi-task learning. The introduction of language vector representations derived from both acoustic and phonetic data is particularly innovative and addresses the challenge of data scarcity in low-resource languages. The authors employ a well-structured experimental setup, using the Common Voice dataset and the Wav2vec 2.0 model as a baseline. The use of hierarchical multi-task learning and auxiliary language embeddings is methodologically sound and well-justified.

Comprehensive Evaluation: The paper provides a thorough evaluation of the proposed methods, including in-domain and out-of-domain scenarios. The results show consistent improvements in PER across different configurations, particularly for low-resource languages, which is a significant achievement.

Practical Implications: The proposed methods have practical implications for linguistic research, language education, and speech therapy, as they enhance the ability of ASR systems to generalize to unseen languages and handle linguistic diversity.

Weaknesses:

Lack of Comparison with State-of-the-Art Models: While the paper compares the proposed methods to a baseline model, it does not provide a comprehensive comparison with other state-of-the-art multilingual ASR models, such as Whisper or other recent transformer-based architectures. This limits the ability to fully assess the relative performance of the proposed approach.

Limited Discussion on Computational Costs: The paper does not discuss the computational costs associated with the proposed methods, particularly the hierarchical multi-task learning approach and the training of language vector representations. This information would be valuable for researchers and practitioners considering the adoption of these methods.

Generalization to Unseen Languages: While the paper demonstrates improvements in out-of-domain scenarios, the extent to which the proposed methods can generalize to entirely unseen languages (zero-shot learning) is not thoroughly explored. More detailed experiments on zero-shot learning would strengthen the paper's claims.

Ablation Studies: The paper would benefit from more detailed ablation studies to isolate the contributions of individual components (e.g., hierarchical multi-task learning vs. language embeddings) to the overall performance improvement.

Suggestions for Improvement:

Comparative Analysis: Include a comparison with other state-of-the-art multilingual ASR models to provide a clearer picture of where the proposed methods stand in relation to existing approaches.

Computational Cost Analysis: Provide an analysis of the computational costs associated with the proposed methods, including training time, memory requirements, and inference speed.

Zero-Shot Learning Experiments: Expand the evaluation to include more detailed experiments on zero-shot learning, particularly for languages that are entirely absent from the training data.

Ablation Studies: Conduct ablation studies to better understand the individual contributions of the hierarchical multi-task learning approach and the language embeddings to the overall performance improvement.

---

### Official Review · Reviewer_U5ar · 2025-02-25
**ADOPTING DOMAIN-SPECIFIC KNOWLEDGE IN ASR SYSTEMS**

**Rating:** 7
**Confidence:** 3

**Review:**

The study is devoted to the issues of improving speech recognition models in the format of the international phonetic alphabet.
It is noted that the purpose of the study is an attempt to overcome the problems of the lack of annotated data corpora for various languages ​​and to improve the accuracy of speech recognition systems.
A model for including auxiliary linguistic information is proposed. Two options for representing language vectors are introduced: one obtained from acoustic signals, and the other from phonetic transcriptions. The results of the study showed that embedding close languages, one linguistic group, improves the quality of the result, and accordingly, worsens it if the languages ​​​​are different groups.
In general, the authors claim that the model improves the accuracy and stability of multilingual systems from 7 to 20%. Probably, in practice, this will be a significant contribution to the development of speech recognition systems.

---

### Official Review · Reviewer_LZuv · 2025-02-27
**The article proposes an innovative approach to improving multilingual automatic speech recognition systems in International Phonetic Alphabet format through the integration of linguistic knowledge and auxiliary information using hierarchical multi-task learning and language vector representations, demonstrating a significant improvement in recognition accuracy for both known and unknown languages.**

**Rating:** 7
**Confidence:** 3

**Review:**

This paper addresses the challenge of enhancing multilingual automatic speech recognition (ASR) systems, particularly for transcribing speech into the International Phonetic Alphabet (IPA). The authors propose two distinct approaches: hierarchical multi-task learning (HMTL) and auxiliary language embeddings derived from both acoustic signals and phonetic transcriptions. These methods aim to improve model generalization across diverse phonetic systems, especially for low-resource languages.

Strengths.

Originality and Significance: The integration of domain-specific linguistic knowledge into the training process is a novel contribution. The use of HMTL to decompose phoneme recognition into multiple abstraction levels and the introduction of auxiliary language embeddings are innovative solutions to the problem of data scarcity and linguistic diversity.

Comprehensive Evaluation: The authors conduct extensive experiments using datasets with varying numbers of languages, demonstrating consistent improvements in Phoneme Error Rate (PER) metrics across in-domain and out-of-domain scenarios. The results show relative improvements of 7–10% for in-domain languages and up to 30% for out-of-domain languages.

Practical Implications: The proposed methods have practical applications in linguistic research, language education, and speech therapy. They also contribute to improving the robustness of ASR systems for unseen languages.

Clarity and Structure: The paper is well-organized, with clear sections dedicated to methodology, experimental setup, and results. The figures and tables effectively illustrate the findings.

Weaknesses.

Limited Scope of Real-World Validation: While the synthetic dataset used for training and validation is well-constructed, the paper lacks real-world testing on noisy or complex data. This limits the ability to fully assess the practical utility of the proposed methods.
Computational Complexity Analysis: The computational cost of the proposed approaches, especially HMTL and metric learning, is not thoroughly analyzed. A discussion of scalability and efficiency would be beneficial for practitioners considering these methods.

Comparison with State-of-the-Art Models: Although the authors compare their approach to a baseline model, there is no comprehensive comparison with other state-of-the-art multilingual ASR systems like Whisper or recent transformer-based architectures. Such comparisons would provide a clearer understanding of the relative performance.

Formatting Issues: Minor typographical errors and inconsistent formatting detract slightly from the overall presentation quality.

Conclusion:
The paper makes a valuable contribution to the field of multilingual ASR by introducing methods that leverage hierarchical learning and auxiliary information. However, addressing the aforementioned weaknesses—especially real-world validation, computational complexity analysis, and broader comparisons—would further enhance its impact.

---

### Author Rebuttal · Authors · 2025-03-06

Limited Scope of Real-World Validation: The choice of synthetic data using g2p algorithms is related to the interesting work in a multilingual problem statement and generalization to new languages, but high-quality manual labeling is unfortunately extremely limited. Also, the detail and quality of such labeling varies greatly between languages ​​and data sets, due to which it would be difficult to obtain unambiguous results with low dispersion on real data. But in general, the observation is really important and in the future it is supposed to collect a sample based on existing data for a number of languages.

Comparison with State-of-the-Art Models: The challenge in comparison lies in the absence of a standardized benchmark for ASR in IPA format. In the earlier works cited, an internal test based on the authors' own data split was used, which, unfortunately, cannot be reconstructed. Therefore, a similar approach was chosen, focusing on comparing the methodologies as such on fixed training data. As for classical ASR models like Whisper, the model is more focused on deeply considering sentence context during recognition and provides output in the specified language. In contrast, this work is more dedicated to accurately recognizing what was actually heard, rather than what was intended.

Generalization to Unseen Languages: The out-of-domain experiments served as a test of the metrics for languages not included in the dataset. In all experiments, this issue was explored for various held-out languages, as shown, for example, in Table 5. What truly deserved consideration in the study were low-resource languages, which were less likely to be overrepresented in the XLSR model's training. However, for such languages, it is challenging to find high-quality annotations, which is why not all obtained results were deemed reliable and were excluded from the final version (experiments were conducted on languages of small indigenous peoples of Russia). As a result, the focus of the work shifted to removing from the training dataset those languages for which high-quality G2P algorithms or manual annotations already existed.

---

### Decision · Program_Chairs · 2025-03-08

**Decision:**

Accept (Oral)

**Comment:**

Your article has been accepted and you can make a presentation on the article. All articles will be sorted by rating and within the available conference places one author from each article will be invited. If there are not enough places, then you will either have the opportunity to present remotely or come at your own expense!